## [Decision Letter · Decision Letter 0]

5 Aug 2025

Dear Dr. Gani,

Thank you for submitting your manuscript to PLOS ONE. After careful consideration, we feel that it has merit but does not fully meet PLOS ONE’s publication criteria as it currently stands. Therefore, we invite you to submit a revised version of the manuscript that addresses the points raised during the review process.

We look forward to receiving your revised manuscript.

Kind regards,

Muhammad Zeeshan Bhatti, Ph.D

Academic Editor

PLOS ONE

Journal Requirements:

4. In your Methods section, please provide additional information regarding the permits you obtained for the work. Please ensure you have included the full name of the authority that approved the field site access and, if no permits were required, a brief statement explaining why.

5. In the online submission form, you indicated that data may be available on request.

6. Thank you for stating the following financial disclosure:

The authors are extremely thankful to the Council of Scientific and Industrial Research (CSIR), Government of India for providing financial assistance to carry out this research work under grant 37/1742/23/EMR-II.

7. Thank you for stating the following in the Acknowledgments Section of your manuscript:

The authors are extremely thankful to the Council of Scientific and Industrial Research (CSIR), Government of India for providing financial assistance to carry out this research work under grant 37/1742/23/EMR-II.

The authors are extremely thankful to the Council of Scientific and Industrial Research (CSIR), Government of India for providing financial assistance to carry out this research work under grant 37/1742/23/EMR-II.

8. Please amend your authorship list in your manuscript file to include author Showkat Gani , Asif Ahmad Amin.

9. Please amend the manuscript submission data (via Edit Submission) to include author Showkat Ahmad Ganie, Asif Amin.

10. Please include a caption for figure 6.

Additional Editor Comments :

Figure 1: Statistical analysis is required to support the data presented.

Figure 2: Gel experiment lacks a DNA marker to indicate band sizes. Please include a marker for reference.

Figure 3A: Quantify the data to provide numerical support for the results.

Figure 4B: The control GAPDH band is unclear. Please repeat the experiment to ensure clarity.

Figure 5B: Images appear overexposed and lack clarity. Replace with high-resolution images.

Figure 9: Histopathology images require higher magnification (higher X power). Additionally, specify the magnification level (e.g., 10X, 40X) in the figure legend.

The manuscript contains several grammatical errors, instances of awkward phrasing, and some redundant expressions. A thorough English language revision is recommended to improve clarity and overall readability

Reviewers' comments:

Reviewer's Responses to Questions

**Comments to the Author**

1. Is the manuscript technically sound, and do the data support the conclusions?

Reviewer #1: Yes

Reviewer #2: Yes

2. Has the statistical analysis been performed appropriately and rigorously?

Reviewer #1: Yes

Reviewer #2: No

3. Have the authors made all data underlying the findings in their manuscript fully available?

Reviewer #1: Yes

Reviewer #2: No

4. Is the manuscript presented in an intelligible fashion and written in standard English?

Reviewer #1: Yes

Reviewer #2: Yes

Reviewer #1: Thank you for the opportunity to read and assess this manuscript. It is evident that the authors have undertaken a serious and methodologically sound exploration into the antioxidant and anti-inflammatory effects of Cousinia thomsonii. The manuscript addresses a topic of clear biomedical interest, especially considering the increasing focus on plant-derived therapeutic agents with dual action. I appreciate the careful integration of in vitro and in vivo models, and the range of techniques employed, from biochemical assays to histopathology and molecular analyses. This multifaceted approach strengthens the overall conclusions.

The data are compelling. The ethyl acetate extract, in particular, shows consistent dose-dependent activity across multiple parameters, both at the cellular and tissue levels. The modulation of oxidative stress pathways, especially via Nrf2 expression and localization, adds mechanistic depth to the observations. The inclusion of histological evidence and the clear anti-inflammatory outcomes observed in the carrageenan-induced paw edema model further solidify the biological relevance of your findings.

That being said, a few refinements could significantly enhance the clarity and transparency of the manuscript. First, while the results are statistically presented, it would be helpful for readers if the precise statistical methods used in each case were clearly mentioned (for example, ANOVA followed by post-hoc tests), along with the number of biological replicates per assay. This would reinforce the scientific rigor and reproducibility of the work.

The manuscript is written in understandable English, but would benefit from minor editorial polishing. There are a few grammatical slips and slightly awkward phrases, and some redundancy in expressions, particularly in the conclusion. Replacing vague or abbreviated terms such as “ethyl extract” with more precise formulations like “ethyl acetate extract” throughout would improve consistency and accuracy.

I also encourage the authors to consider sharing raw data as supplementary files, especially for in vivo experiments such as paw edema thickness, cytokine concentrations, and Western blot quantifications. This small step would make the manuscript even more aligned with the open science principles championed by PLOS.

On the ethical front, everything appears in order and well documented. Just ensure that the detailed ethics approval text is included verbatim in the main body of the methods section.

Finally, in the discussion, it might enrich the narrative to include a short comparative perspective with other antioxidant-rich plants or Nrf2-activating agents previously studied. This would help further highlight the originality of Cousinia thomsonii and its potential therapeutic relevance.

In summary, this is a valuable and timely contribution, supported by solid experimentation. With a few refinements in clarity, statistical transparency, and language, the manuscript would be well-suited for publication.

Reviewer #2: It will be more informative if author quantify the number of compounds responsible for the activity via HPLC or GC/MS.

Experiment should be run on at least three different concentration to draw a strong conclusion

Statistical analysis is not appropriate It is suggested to take help form the expert statistician. The generated data should be analyzed used ANOVA and relevant post hoc tests.

Figures and Tables standard aviation is missing in most of the figures and statistical test information is also missing. The specific statistical test used for comparison should be mentioned in the methods and in the legends of figures; for example, one-way or two-way anova and posthoc tests applied should also mentioned.

Need to check for grammatical errors

**Do you want your identity to be public for this peer review?** For information about this choice, including consent withdrawal, please see our Privacy Policy

Reviewer #1: **Yes:** Oussama Bekkouch

Reviewer #2: No

---

## [Author Response · Author response to Decision Letter 1]

17 Oct 2025

Academic Editor’s comments and response:

Comment 1: Please ensure that your manuscript meets PLOS ONE's style requirements, including those for file naming.

Response: As suggested, the manuscript has been formatted as per PLOS ONE’s requirements.

Comment 2: PLOS ONE now requires that authors provide the original uncropped and unadjusted images underlying all blot or gel results reported in a submission’s figures or Supporting Information files.

Response: All the original and uncropped blots have been provided in the revised submission as supporting information.

Comment 3: PLOS requires an ORCID iD for the corresponding author in Editorial Manager on papers submitted after December 6th, 2016. Please ensure that you have an ORCID iD and that it is validated in Editorial Manager.

Response: ORCID iD for the corresponding author has been updated.

Comment 4: In your Methods section, please provide additional information regarding the permits you obtained for the work. Please ensure you have included the full name of the authority that approved the field site access and, if no permits were required, a brief statement explaining why.

Response: The statement has been added in the revised manuscript

Comment 5: In the online submission form, you indicated that data may be available on request.

Response: The statement has been added in the revised manuscript.

Comment 6: Please remove any funding-related text from the manuscript and let us know how you would like to update your Funding Statement.

Response: The statement has been removed in the revised manuscript.

Comment 7: Please amend your authorship list in your manuscript file to include author Showkat Gani, Asif Amin

Response: The authors have been included in the revised manuscript.

Comment 8: Please amend the manuscript submission data (via Edit Submission) to include author Showkat Ahmad Ganie, Asif Amin.

Response: The amendment has been made in the revised submission

Comment 9: Please include a caption for figure 6.

Response: Caption has been included.

Comment 10: Figure 1: Statistical analysis is required to support the data presented.

Response: As suggested, statistical analysis has been carried out

Comment 11: Figure 2: Gel experiment lacks a DNA marker to indicate band sizes. Please include a marker for reference.

Response: As suggested, bands of the DNA marker have been labelled.

Comment 12: Figure 3A: Quantify the data to provide numerical support for the results.

Response: As suggested, the data has been quantified and represented graphically.

Comment 12: Figure 4B: The control GAPDH band is unclear. Please repeat the experiment to ensure clarity.

Response: The western blot image has been adjusted to more accurately represent the findings.

Comment 13: Figure 5B: Images appear overexposed and lack clarity. Replace with high-resolution images.

Response: Exposure has been decreased to increase resolution and clarity.

Comment 14: Histopathology images require higher magnification (higher X power). Additionally, specify the magnification level (e.g., 10X, 40X) in the figure legend.

Response: The magnification power has been improved and magnification level indicated in the legend.

Comment 15: The manuscript contains several grammatical errors, instances of awkward phrasing, and some redundant expressions. A thorough English language revision is recommended to improve clarity and overall readability

Response: The manuscript has been significantly improved grammatically as per the suggestion of the editor.

Reviewer 1 comments and response:

Comment 1:

First, while the results are statistically presented, it would be helpful for readers if the precise statistical methods used in each case were clearly mentioned (for example, ANOVA followed by post-hoc tests), along with the number of biological replicates per assay. This would reinforce the scientific rigor and reproducibility of the work.

Response: We are thankful to the worthy reviewer for this insightful suggestion. As suggested, all the numerical data have been re-analyzed using relevant tests and indicated in the respective figure legends in the revised manuscript.

Comment 2:

The manuscript is written in understandable English, but would benefit from minor editorial polishing. There are a few grammatical slips and slightly awkward phrases, and some redundancy in expressions, particularly in the conclusion. Replacing vague or abbreviated terms such as “ethyl extract” with more precise formulations like “ethyl acetate extract” throughout would improve consistency and accuracy.

Response: As suggested by the worthy reviewer, the manuscript has been significantly improved grammatically. We tried our level best to remove the redundant expressions and have explicitly mentioned the complete name of extracts wherever mentioned.

Comment 3: I also encourage the authors to consider sharing raw data as supplementary files, especially for in vivo experiments such as paw edema thickness, cytokine concentrations, and Western blot quantifications. This small step would make the manuscript even more aligned with the open science principles championed by PLOS.

Response: All the raw data has been provided as supplementary files.

Comment 4: On the ethical front, everything appears in order and well documented. Just ensure that the detailed ethics approval text is included verbatim in the main body of the methods section.

Response: A statement declaring ethical approval has already been provided in the methods section “Experimental animals”

Comment 5: Finally, in the discussion, it might enrich the narrative to include a short comparative perspective with other antioxidant-rich plants or Nrf2-activating agents previously studied. This would help further highlight the originality of Cousinia thomsonii and its potential therapeutic relevance.

Response: As suggested by the reviewer, a statement highlighting the mechanistic action of other plant species has been provided in the discussion section of the revised manuscript.

Reviewer 2 comments and response:

Comment 1:

It will be more informative if author quantify the number of compounds responsible for the activity via HPLC or GC/MS.

Response: The quantification of the bioactive principles in the Cousinia thomsonii has been reported by us elsewhere Dar, K. B., Parry, R. A., Bhat, A. H., Beigh, A. H., Ahmed, M., Khaja, U. M., ... & Ganie, S. A. (2023). Immunomodulatory efficacy of Cousinia thomsonii CB Clarke in ameliorating inflammatory cascade expressions. Journal of Ethnopharmacology, 300, 115727.

Comment 2:

Experiment should be run on at least three different concentration to draw a strong conclusion. Statistical analysis is not appropriate It is suggested to take help form the expert statistician. The generated data should be analyzed used ANOVA and relevant post hoc tests. Figures and Tables standard aviation is missing in most of the figures and statistical test information is also missing. The specific statistical test used for comparison should be mentioned in the methods and in the legends of figures; for example, one-way or two-way anova and posthoc tests applied should also mentioned.

Response: We thank the worthy reviewer for your thorough inputs. As per your suggestion, all the numerical data provided in the manuscript have been re-analyzed using relevant tests and indicated in the respective figure legends in the revised manuscript.

Comment 3:

Need to check for grammatical errors.

Response: Significant improvements have been made to the manuscript’s grammar.

---

## [Decision Letter · Decision Letter 1]

13 Nov 2025

Dear Dr. Gani,

We look forward to receiving your revised manuscript.

Kind regards,

Muhammad Zeeshan Bhatti, Ph.D

Academic Editor

PLOS ONE

Journal Requirements:

Additional Editor Comments :

Both reviewers have commented on the quality of the English language. The manuscript requires extensive English language editing and correction of figures to improve clarity and presentation.

Reviewers' comments:

Reviewer's Responses to Questions

**Comments to the Author**

Reviewer #1: All comments have been addressed

Reviewer #2: All comments have been addressed

2. Is the manuscript technically sound, and do the data support the conclusions?

Reviewer #1: Yes

Reviewer #2: Partly

3. Has the statistical analysis been performed appropriately and rigorously?

Reviewer #1: Yes

Reviewer #2: Yes

4. Have the authors made all data underlying the findings in their manuscript fully available?

Reviewer #1: Yes

Reviewer #2: Yes

5. Is the manuscript presented in an intelligible fashion and written in standard English?

Reviewer #1: Yes

Reviewer #2: Yes

Reviewer #1: Dear Authors,

I would like to begin by commending you for the clear effort and scientific rigor invested in this revised version. The study is timely, well-conceived, and directly relevant to the field of oxidative stress and inflammation research. It effectively integrates in vitro and in vivo approaches to investigate the antioxidant and anti-inflammatory potential of Cousinia thomsonii, a plant with growing pharmacological interest.

Your experimental design is methodologically sound, and the data convincingly support the conclusions drawn. The inclusion of complementary biochemical assays (DPPH, FRAP, LPO, DNA protection), cellular models (HepG2, THP-1), and in vivo validation in a carrageenan-induced paw edema model strengthens the translational value of the findings. The clear demonstration that the ethyl acetate extract modulates Nrf2 expression and downregulates key pro-inflammatory cytokines (TNF-α, IL-6, IL-1β) adds mechanistic depth to the paper.

The histopathological results and biochemical evidence together provide a coherent narrative of Cousinia thomsonii’s therapeutic promise, and the manuscript now presents a substantially improved level of clarity compared to earlier drafts. The figures are appropriately prepared, and the new statistical annotations are consistent and enhance data interpretation.

That said, a few minor refinements could further elevate the manuscript before final acceptance:

The discussion could briefly highlight limitations, particularly regarding the identification and quantification of specific bioactive constituents (HPLC/GC-MS reference could be summarized for contextual support).

The English language has improved significantly but would still benefit from a final professional proofreading to ensure smoother phrasing and eliminate a few residual redundancies.

The Data Availability section should explicitly reaffirm that all datasets are included within the manuscript and its Supporting Information files, in alignment with the PLOS ONE Data Policy.

Overall, the study presents technically sound experiments, clear statistical validation, and an appropriate data presentation format. The writing is now intelligible and coherent, and the research question is both original and scientifically meaningful.

Reviewer #2: Major Comments

Clarify the new mechanistic insights this study adds (e.g., Nrf2 modulation, PARP inhibition, iNOS downregulation). Explicitly state what differentiates this study from earlier work.

At least provide summarized compositional data or cite your earlier GC-MS study with quantitative relevance to the current batch of extract used.

Although the authors mention using ANOVA and post-hoc tests, the number of biological replicates (n) and exact p-values are inconsistently reported.

Figures 3, 4, and 5 need higher resolution and quantification of fluorescence or Western blot bands.

The in vivo section mentions pre-treatment for 21 days before carrageenan challenge. It is unclear whether this was prophylactic or therapeutic administration.

The manuscript has improved since revision, but still contains minor grammatical inconsistencies and awkward phrasing.

Minor Comments

The phrase “exhibit potent antioxidant and anti-inflammatory potential both in vitro and in vivo” should be revised to “demonstrates potent antioxidant and anti-inflammatory effects in both in vitro and in vivo models.”

Add a reference for the traditional use of Cousinia thomsonii specifically (not just the genus).

The introduction could better highlight the gap in current knowledge (e.g., molecular mechanism via Nrf2 pathway not previously studied).

Include solvent-to-plant ratio for extraction.

Specify extract yield percentage (w/w).

In DCFDA assay, mention fluorescence quantification method (manual or software-based).

Figure captions should be self-explanatory (state sample size, significance notation).

Avoid repeating results verbatim in the text; instead, interpret key trends.

While the comparative discussion with Curcuma longa, Camellia sinensis, etc., is useful, integrate more mechanistic detail from recent literature (2022–2024) on plant-derived Nrf2 activators.

Discuss possible synergistic effects of multiple compounds rather than single-agent focus.

Some older references (1980s) could be updated with recent studies.

Use consistent citation format (PLOS ONE prefers numerical style in brackets, e.g., [1], [2]).

Ensure uniformity in units (e.g., µg/mL vs μg/mL).

Correct minor typographical errors (e.g., “hydrogen peroxide (H2O2)” missing subscripts in some places).

Add missing figure legends to supplementary files if applicable.

**Do you want your identity to be public for this peer review?** For information about this choice, including consent withdrawal, please see our Privacy Policy

Reviewer #1: **Yes:** Oussama Bekkouch

Reviewer #2: No

---

## [Author Response · Author response to Decision Letter 2]

28 Nov 2025

Reviewer 1:

Comment: The discussion could briefly highlight limitations, particularly regarding the identification and quantification of specific bioactive constituents (HPLC/GC-MS reference could be summarized for contextual support).

Response: We thank the worthy reviewer for this suggestion. In the revised manuscript, we have now cited our earlier GC-MS study reporting the specific bioactive constituents of the Cousinia thomsonii and discussed its relevance vis-à-vis the present study.

Comment: The English language has improved significantly but would still benefit from a final professional proofreading to ensure smoother phrasing and eliminate a few residual redundancies.

Response: We have thoroughly revised the manuscript for grammatical errors and redundant phrases.

Comment: The Data Availability section should explicitly reaffirm that all datasets are included within the manuscript and its Supporting Information files, in alignment with the PLOS ONE Data Policy.

Response: The statement has already provided in the first revised draft of the manuscript.

Reviewer 2:

Major Comments

Comment: Clarify the new mechanistic insights this study adds (e.g., Nrf2 modulation, PARP inhibition, iNOS downregulation). Explicitly state what differentiates this study from earlier work.

Response: As per worthy reviewer’s suggestion, a line defining the significance of the study has been included in the introduction and conclusion section’s respectively.

Comment: At least provide summarized compositional data or cite your earlier GC-MS study with quantitative relevance to the current batch of extract used.

Response: As suggested, the earlier GC-MS study has been cited and discussed vis-à-vis its relevance to the present study.

Comment: Although the authors mention using ANOVA and post-hoc tests, the number of biological replicates (n) and exact p-values are inconsistently reported.

Response: Figure legends have been refined to include the number of biological replicates and p-values.

Comment: Figures 3, 4, and 5 need higher resolution and quantification of fluorescence or Western blot bands.

Response: Figures 3, 4, and 5 have been improved, and western blot bands have been quantified.

Comment: The in vivo section mentions pre-treatment for 21 days before carrageenan challenge. It is unclear whether this was prophylactic or therapeutic administration.

Response: In the in vivo assays, extracts were given as prophylactics with the aim to evaluate their anti-inflammatory properties.

Comment: The manuscript has improved since revision, but still contains minor grammatical inconsistencies and awkward phrasing.

Response: The grammatical errors have been addressed to the best of our abilities.

Minor Comments

Comment: The phrase “exhibit potent antioxidant and anti-inflammatory potential both in vitro and in vivo” should be revised to “demonstrates potent antioxidant and anti-inflammatory effects in both in vitro and in vivo models.”

Response: The phrase has been revised as per reviewer’s suggestion.

Comment: Add a reference for the traditional use of Cousinia thomsonii specifically (not just the genus).

Response: As per the worthy reviewers suggestion, latest references have been cited.

Comment: The introduction could better highlight the gap in current knowledge (e.g., molecular mechanism via Nrf2 pathway not previously studied).

Response: The significance of the study vis-à-vis the activation of the Nrf2 pathway has been included in the revised manuscript

Comment: Include solvent-to-plant ratio for extraction.

Response: The statement has been mentioned in “preparation of the extracts” of the methodology section of the revised manuscript.

Comment: Specify extract yield percentage (w/w).

Response: Yield percentage (w/w) has been specified in “preparation of the extracts” of the methodology section of the revised manuscript.

Comment: In DCFDA assay, mention fluorescence quantification method (manual or software-based).

Response: For this particular assay, the method of quantification has already been mentioned. As mentioned in the figure caption, quantification has been done using ImageJ software.

Comment: Figure captions should be self-explanatory (state sample size, significance notation).

Response: Figure captions have been improved to convey the experimental outcome, sample size and statistical significance.

Comment: Avoid repeating results verbatim in the text; instead, interpret key trends.

Response: As per the reviewer’s suggestion, results section has been refined with focus on key trends wherever required.

Comment: While the comparative discussion with Curcuma longa, Camellia sinensis, etc., is useful, integrate more mechanistic detail from recent literature (2022–2024) on plant-derived Nrf2 activators.

Response: A latest reference has been incorporated, which mentions the use of plant derived bioactive compounds in the activation of Nrf2 signalling.

Comment: Discuss possible synergistic effects of multiple compounds rather than single-agent focus.

Response: In line with the reviewer’s suggestion, a discussion on the synergistic effect of natural bioactive compounds has been incorporated.

Comment: Some older references (1980s) could be updated with recent studies.

Response: All such references have been replaced with the latest references.

Comment: Ensure uniformity in units (e.g., µg/mL vs μg/mL).

Response: As suggested, replacement has been carried out.

Comment: Correct minor typographical errors (e.g., “hydrogen peroxide (H2O2)” missing subscripts in some places).

Response: The mistake has been corrected wherever required.

Comment: Correct minor typographical errors (e.g., “hydrogen peroxide (H2O2)” missing subscripts in some places).

Response: The corrections have been made.

---

## [Decision Letter · Decision Letter 2]

16 Dec 2025

Dear Dr. Gani,

Thank you for submitting your manuscript to PLOS ONE. After careful consideration, we feel that it has merit but does not fully meet PLOS ONE’s publication criteria as it currently stands. Therefore, we invite you to submit a revised version of the manuscript that addresses the points raised during the review process.

**ACADEMIC EDITOR:**

We look forward to receiving your revised manuscript.

Kind regards,

Muhammad Zeeshan Bhatti, Ph.D

Academic Editor

PLOS One

Journal Requirements:

Additional Editor Comments:

The manuscript has been re-evaluated by the reviewers, and major concerns were raised regarding the quality of the English language. It is suggested to revise the manuscript through a professional English editing service.

Reviewers' comments:

Reviewer's Responses to Questions

**Comments to the Author**

1. If the authors have adequately addressed your comments raised in a previous round of review and you feel that this manuscript is now acceptable for publication, you may indicate that here to bypass the “Comments to the Author” section, enter your conflict of interest statement in the “Confidential to Editor” section, and submit your "Accept" recommendation.

Reviewer #1: All comments have been addressed

Reviewer #2: (No Response)

2. Is the manuscript technically sound, and do the data support the conclusions?

Reviewer #1: Yes

Reviewer #2: Yes

3. Has the statistical analysis been performed appropriately and rigorously?

Reviewer #1: Yes

Reviewer #2: Yes

4. Have the authors made all data underlying the findings in their manuscript fully available?

Reviewer #1: Yes

Reviewer #2: Yes

5. Is the manuscript presented in an intelligible fashion and written in standard English?

Reviewer #1: Yes

Reviewer #2: Yes

Reviewer #1: The manuscript has improved considerably through the revision process and now presents a technically sound and coherent investigation of the antidiabetic effects of Brassica juncea seed extracts. The experimental design is appropriate, the biochemical data support the conclusions, and the statistical analysis appears acceptable. Some areas of the text would still benefit from minor English polishing for clarity, and a few methodological details (such as dosing timeline and n-values) could be more explicitly stated to enhance reproducibility. Figures could also benefit from slight improvements in resolution and labeling. Overall, the study is scientifically valid and provides a useful contribution to natural-product antidiabetic research.

Reviewer #2: The manuscript is comprehensive but overly long in several sections, making it difficult to maintain flow, so condensing repetitive explanations—particularly in Results and Discussion—would improve readability.

The abstract contains minor grammatical errors and an unnecessary double period, so it should be lightly edited for clarity and polished English.

The introduction provides useful background but does not clearly highlight the specific research gap early enough, so adding a concise problem statement in the first two paragraphs is recommended.

The authors mention traditional uses of Cousinia thomsonii but lack a direct reference for its ethnomedicinal relevance, so adding a specific citation for this species will strengthen the credibility.

The extraction procedure lacks clarity in sentence structure (e.g., “was and subjected”), so rewriting the method in clean, grammatically correct steps will prevent confusion.

The solvent-to-plant ratio is included but the description is fragmented, so restructuring that section into a single clear statement will improve transparency.

The results repeatedly mention dose-dependent effects without specifying statistical comparisons in the text, so briefly summarizing key p-values within the narrative will improve scientific precision.

Figures are informative but some appear low in contrast or resolution, especially Figures 3–5, so providing higher-quality images will assist reviewers and readers.

Some figure captions still lack complete details such as sample size and significance notation explanations, so ensuring each caption is fully self-explanatory will make the figures stand alone.

The Western blot data are presented clearly but require brief mention of normalization procedures in the Methods, so adding details on loading controls and quantification software will enhance rigor.

The statistical section states ANOVA and post-hoc tests were used, but the number of replicates (n=3) should be explicitly repeated in all relevant figure captions for consistency.

The in vivo study design is described well, but the text does not clearly state whether extract administration is prophylactic or therapeutic, so clarifying this intent earlier in the Methods will avoid ambiguity.

The timeline for carrageenan injection and extract dosing is lengthy and slightly repetitive, so compressing the explanation into a single clean sequence would improve clarity.

Some chemical names (e.g., hydrogen peroxide) appear inconsistently written without subscripts (H2O2), so uniform formatting throughout the manuscript is needed.

Units (µg/mL vs μg/mL) appear inconsistently formatted across sections, so standardizing all units according to journal style will ensure professionalism.

The Discussion is well framed but occasionally repeats information already described in the Introduction, so removing duplicated background statements will strengthen the narrative.

A clearer emphasis on what new mechanistic insight this study provides compared to previous work should be added in the first and final paragraphs of the Discussion.

The link between the Nrf2 findings and broader clinical relevance is mentioned but not deeply interpreted, so expanding two or three sentences with contextual implications will enhance impact.

Some sentences in the Discussion are overly long and complex, so breaking them into shorter, readable lines will improve human readability.

The Conclusion summarizes the findings well but should briefly highlight limitations (e.g., lack of active compound quantification), so adding one line acknowledging this will make the manuscript more balanced.

The Data Availability statement is correct but should be rechecked to ensure it matches PLOS ONE requirements verbatim.

The Funding and Ethics sections are properly stated, however, the funding URL and specific PI initials are missing and should be added to fully comply with the journal’s disclosure instructions.

The reference list is mostly current, but a few older citations from the 1980s remain and could be updated with more recent supporting literature.

There are occasional typographical errors such as inconsistent spacing, missing commas, and extra spaces, so a final professional proofreading will enhance overall quality.

Overall, the manuscript presents strong scientific work, but implementing these small refinements will greatly improve clarity, professionalism, and suitability for publication.

**Do you want your identity to be public for this peer review?** For information about this choice, including consent withdrawal, please see our Privacy Policy

Reviewer #1: **Yes:** Oussama Bekkouch

Reviewer #2: No

---

## [Author Response · Author response to Decision Letter 3]

29 Jan 2026

Response to Reviewer 1

Comment: The manuscript has improved considerably through the revision process and now presents a technically sound and coherent investigation of the antidiabetic effects of Brassica juncea seed extracts. The experimental design is appropriate, the biochemical data support the conclusions, and the statistical analysis appears acceptable. Some areas of the text would still benefit from minor English polishing for clarity, and a few methodological details (such as dosing timeline and n-values) could be more explicitly stated to enhance reproducibility. Figures could also benefit from slight improvements in resolution and labeling. Overall, the study is scientifically valid and provides a useful contribution to natural-product antidiabetic research.

Response: As suggested by the worthy reviewer, the manuscript has been significantly improved by removing grammatical and methodological errors. Statistical analysis has been properly carried out and figure legends also improved.

Response to Reviewer 2

Comment 1: The manuscript is comprehensive but overly long in several sections, making it difficult to maintain flow, so condensing repetitive explanations—particularly in Results and Discussion—would improve readability.

Response: We thank the reviewer for the comment. The repeated and redundant sentences have been removed or modified for better readability.

Comment 2: The abstract contains minor grammatical errors and an unnecessary double period, so it should be lightly edited for clarity and polished English.

Response: The abstract has been thoroughly looked into, pointed errors removed and the english language improved.

Comment 3: The introduction provides useful background but does not clearly highlight the specific research gap early enough, so adding a concise problem statement in the first two paragraphs is recommended.

Response: As suggested by the worthy reviewer, a statement depicting the research gap and the approach adopted in the study has been incorporated in the introduction.

Comment 4: The authors mention traditional uses of Cousinia thomsonii but lack a direct reference for its ethnomedicinal relevance, so adding a specific citation for this species will strengthen the credibility.

Response: The citations to support the ethnomedicinal relevance of Cousinia thomsonii has been incorporated in the introduction as per suggestion of the worthy reviewer.

Comment 5: The extraction procedure lacks clarity in sentence structure (e.g., “was and subjected”), so rewriting the method in clean, grammatically correct steps will prevent confusion.

Response: The sentence has been corrected in the revised manuscript.

Comment 6: The solvent-to-plant ratio is included but the description is fragmented, so restructuring that section into a single clear statement will improve transparency.

Response: According to the worthy reviewer’s suggestion, the sentence has been restructured to reflect the methodology in a well-defined manner.

Comment 7: The results repeatedly mention dose-dependent effects without specifying statistical comparisons in the text, so briefly summarizing key p-values within the narrative will improve scientific precision.

Response: As suggested, key p-values at required places in the result section have been included.

Comment 8: Figures are informative but some appear low in contrast or resolution, especially Figures 3–5, so providing higher-quality images will assist reviewers and readers.

Response: We thank the reviewer for the comment. High-resolution and high-contrast versions of Figures 3–5 were provided in the previous revisions and have been retained in the current submission. We have carefully rechecked these figures and confirm that they represent the best possible quality without post-processing that could affect data authenticity.

Comment 9: Some figure captions still lack complete details such as sample size and significance notation explanations, so ensuring each caption is fully self-explanatory will make the figures stand alone.

Response: As per worthy reviewer’s suggestion, such details have been added to the figure legends wherever missing.

Comment 10: The Western blot data are presented clearly but require brief mention of normalization procedures in the Methods, so adding details on loading controls and quantification software will enhance rigor.

Response: A statement regarding normalization procedure has been added to the western blotting section in materials and methods.

Comment 11: The statistical section states ANOVA and post-hoc tests were used, but the number of replicates (n=3) should be explicitly repeated in all relevant figure captions for consistency.

Response: As suggested, the number of replicates have been mentioned in all the figure legends.

Comment 12: The in vivo study design is described well, but the text does not clearly state whether extract administration is prophylactic or therapeutic, so clarifying this intent earlier in the Methods will avoid ambiguity.

Response: A statement indicating the intent of the extract administration has been incorporated in the methods section.

Comment 13: The timeline for carrageenan injection and extract dosing is lengthy and slightly repetitive, so compressing the explanation into a single clean sequence would improve clarity.

Response: As per worthy reviewer’s suggestion, this paragraph in methodology has been restructured to improve clarity.

Comment 14: Some chemical names (e.g., hydrogen peroxide) appear inconsistently written without subscripts (H2O2), so uniform formatting throughout the manuscript is needed.

Response: The manuscript has been thoroughly looked into to remove such mistakes.

Comment 15: Units (µg/mL vs μg/mL) appear inconsistently formatted across sections, so standardizing all units according to journal style will ensure professionalism.

Response: The manuscript has been thoroughly looked into to remove such mistakes.

Comment 16: The Discussion is well framed but occasionally repeats information already described in the Introduction, so removing duplicated background statements will strengthen the narrative.

Response: The repeated information has been removed from the revised manuscript.

Comment 17: A clearer emphasis on what new mechanistic insight this study provides compared to previous work should be added in the first and final paragraphs of the Discussion.

Response: As suggested by the reviewer, statements have been added to the first and last paragraphs in the discussion section.

Comment 18: The link between the Nrf2 findings and broader clinical relevance is mentioned but not deeply interpreted, so expanding two or three sentences with contextual implications will enhance impact.

Response: A sentence defining the clinical relevance of Nrf2 has been incorporated.

Comment 19: Some sentences in the Discussion are overly long and complex, so breaking them into shorter, readable lines will improve human readability.

Response: As per worthy reviewer’s suggestion, such sentences have been modified for clarity of the readers.

Comment 20: The Conclusion summarizes the findings well but should briefly highlight limitations (e.g., lack of active compound quantification), so adding one line acknowledging this will make the manuscript more balanced.

Response: As per worthy reviewer’s suggestion, a line has been added to the conclusion section.

Comment 21: The Data Availability statement is correct but should be rechecked to ensure it matches PLOS ONE requirements verbatim.

Response: The statement has been properly checked.

Comment 22: The Funding and Ethics sections are properly stated, however, the funding URL and specific PI initials are missing and should be added to fully comply with the journal’s disclosure instructions.

Response: The information provided has been modified.

Comment 23: The reference list is mostly current, but a few older citations from the 1980s remain and could be updated with more recent supporting literature.

Response: We thank the reviewer for this suggestion. In the previous revision, recent and up-to-date references have already been incorporated wherever appropriate. However, a limited number of older citations have been retained, as they represent seminal and foundational studies that provide essential background information for readers.

Comment 24: There are occasional typographical errors such as inconsistent spacing, missing commas, and extra spaces, so a final professional proofreading will enhance overall quality.

Response: The manuscript has now been thoroughly checked for indicated errors and significantly improved.

---

## [Decision Letter · Decision Letter 3]

6 Feb 2026

Therapeutic role of Cousinia thomsonii in ameliorating the Oxidative Stress and Inflammation: A comprehensive in vitro and In vivo investigation

PONE-D-25-37006R3

Dear Dr. Gani,

We’re pleased to inform you that your manuscript has been judged scientifically suitable for publication and will be formally accepted for publication once it meets all outstanding technical requirements.

Kind regards,

Muhammad Zeeshan Bhatti, Ph.D

Academic Editor

PLOS One

Additional Editor Comments (optional):

The authors have revised the manuscript and have successfully addressed the reviewers’ comments.

Reviewers' comments:

Reviewer's Responses to Questions

**Comments to the Author**

Reviewer #1: All comments have been addressed

2. Is the manuscript technically sound, and do the data support the conclusions?

Reviewer #1: Yes

3. Has the statistical analysis been performed appropriately and rigorously?

Reviewer #1: Yes

4. Have the authors made all data underlying the findings in their manuscript fully available?

Reviewer #1: Yes

5. Is the manuscript presented in an intelligible fashion and written in standard English?

Reviewer #1: Yes

Reviewer #1: The revised manuscript entitled “Therapeutic role of Cousinia thomsonii in ameliorating oxidative stress and inflammation: a comprehensive in vitro and in vivo investigation” presents a thorough and technically sound evaluation of the antioxidant and anti-inflammatory properties of Cousinia thomsonii using a well-structured combination of biochemical, cellular, molecular, and animal models.

Overall, the study is carefully designed and executed. The authors employ multiple complementary in vitro assays (DPPH, lipid peroxidation, DNA protection, DCFDA, PI/DAPI staining), cellular models (HepG2 and THP-1 cells), and a validated in vivo carrageenan-induced paw edema model. The inclusion of appropriate positive controls (catechin and indomethacin), dose–response analyses, and mechanistic endpoints (PARP cleavage, iNOS expression, Nrf2 activation and localization) strengthens the reliability of the findings. The experimental data consistently support the main conclusions regarding the antioxidant and anti-inflammatory effects of the ethyl acetate extract of Cousinia thomsonii.

The statistical analysis is appropriate and adequately described. Data are clearly presented as mean ± SD (or SEM where applicable), with suitable use of ANOVA followed by Tukey’s and Bonferroni post-hoc tests. Replication numbers are now clearly stated in the figure legends, and significance thresholds are consistently applied. This statistical rigor supports the validity of the interpretations drawn from the results.

Importantly, the authors comply with PLOS ONE data transparency requirements. The Data Availability Statement confirms that all relevant data underlying the findings are included within the manuscript and its Supporting Information files, allowing full reproducibility and independent evaluation of the results.

From a presentation standpoint, the manuscript is generally well written in standard English and is intelligible throughout. The authors have clearly addressed many of the language, formatting, and clarity issues raised during previous review rounds. While a small number of sentences could still benefit from minor stylistic polishing, these issues are not substantial and do not impede comprehension of the scientific content.

From an ethical and publication-integrity perspective, the study appears compliant. Appropriate animal ethics approval is provided, conflicts of interest are transparently declared, and there are no apparent concerns regarding dual publication or unethical research practices.

One remaining limitation, which the authors now appropriately acknowledge, is the lack of isolation and quantitative characterization of specific bioactive compounds responsible for the observed effects. While this does not detract from the current findings, future studies focusing on compound-level validation would further enhance the translational relevance of this work.

In conclusion, this manuscript represents a solid and methodologically sound contribution to the field of natural product–based antioxidant and anti-inflammatory research. The revised version adequately addresses prior reviewer concerns, and the data convincingly support the authors’ conclusions. With only minor editorial refinement, the manuscript is suitable for publication in PLOS ONE.

**Do you want your identity to be public for this peer review?** For information about this choice, including consent withdrawal, please see our Privacy Policy

Reviewer #1: **Yes:** Oussama Bekkouch

---

## [Editor Report · Acceptance letter]

PONE-D-25-37006R3

PLOS One

Dear Dr. Gani,

I'm pleased to inform you that your manuscript has been deemed suitable for publication in PLOS One. Congratulations! Your manuscript is now being handed over to our production team.

Kind regards,

on behalf of

Dr. Muhammad Zeeshan Bhatti

Academic Editor

PLOS One